# Factors Influencing the Formalization of Rural Land Transactions in Ethiopia: A Theory of Planned Behavior Approach

**Shewakena Aytenfisu Abab \*, Feyera Senbeta Wakjira and Tamirat Tefera Negash**

Center for Environment and Development, College of Development Studies, Addis Ababa University, Addis Ababa P.O. Box 1176, Ethiopia; feyera.senbeta@aau.edu.et (F.S.W.); tamirat.tefera@aau.edu.et (T.T.N.)
\* Correspondence: shewakena.aytenfisu@aau.edu.et

**Abstract:** Despite the recent successful establishment of systematic land registration programs in some African countries including Ethiopia, updating the land registers has become a growing concern. However, there is limited empirical evidence about whether landholders' behavior is driving the lack of updating land registers in Ethiopia. Using the theory of planned behavior, this study examines the factors that influence landholders' behavior of formalizing rural land transactions in Ethiopia. Primary and secondary data were collected using surveys, key informant interviews, and a literature review. A total of 206 respondents participated in the survey from the Basona Worena district of the Amhara region, central Ethiopia. A structural equation model and descriptive statistics were used to analyze the survey data and supplemented by qualitative findings. The study findings revealed that landholders' attitudes and subjective norms have positively and significantly influenced their intentions to formalize land transactions. However, perceived behavioral control has a negative and insignificant influence. The predictive relevance of the research model is significant and indicates strong intentions to formalize but less actual behavior. This behavior can influence the currency of the information in the land register in the near future and degrade the functions and sustainability of the land registration system in Ethiopia. The study findings recommended facilitating the behavioral changes of landholders to transform their strong intentions into actual practice. Policymakers should develop and implement an innovative information value creation strategy including landholder-oriented services that incentivize the formalization of land transactions and helps landholders overcome hurdles created by subjective norms.

**Keywords:** land register; updating; attitudes; subjective norms; perceived behavioral control; intentions; actual behavior; Ethiopia

## 1. Introduction

Conventionally, land administration systems were created to record and disseminate information about people's relationships to land including rights, responsibilities, restrictions, location and boundaries, and other attributes of real properties [1]. The establishment of land information could take place through either systematic or sporadic land registration approaches [2]. The approach depends on the purpose, available technologies, and source of funding [3,4]. Systematic land registration is commonly compulsory and driven by the state (supply-side), whereas sporadic registration can either be voluntary if initiated by a landholder (demand-side), or compulsory (during a land transfer) [5,6]. The existence of good and well-functioning land information is key to answering fundamental land development and management decision-making questions (i.e., Who has rights? To what land? Where is the land? When was it acquired? How is it used? etc.,) [7]. A land information system (LIS) should be complete, accurate, reliable, uniform, up-to-date, sustainable, and mirror the reality on the ground in real time [8]. Land registration becomes socially more valuable as more parcels are registered, because it leads to more investment and

more transactions [9]. However, the cost of leaving land that is outside of the registry also becomes higher as more land is registered, because investment is lowest for land held under an informal regime [10]. Whether implicit or explicit, the landholder's intention or choices to register one's land and property rights is the result of a reasoned decision [4,10,11]. Therefore, the intentions to formalize land transactions after first-time land certifications are central to the discussion of a functional and sustainable land registration system.

Recently, however, keeping the LIS up-to-date is becoming more challenging than establishing a LIS for the first-time [3,12]. It has also understood that the maintenance and updating of a LIS depends on numerous factors including governance, technology, and behavioral factors [3,12–14]. The latter specifically covers both people working in the system (supply-side capacity) [4,8], and the intent and behavior of land rights holders (demand-side) [2] of formalizing land transactions after first-time land registration. Understanding the factors influencing landholders' land transactions formalization intentions and actual practice, which affects the functionality and sustainability of a LIS are highly important. Without the full cooperation of land rights holders to formalize land rights transactions, the information in the land register could quickly become outdated, inaccurate, unreliable, and less relevant, and the system may rollback into informality and insecurity of tenure [2,3,12,15]. That would be like the previous situation, prior to first-time land registration, and even worse [3].

In Ethiopia, the first level land certification (FLLC) was initiated and implemented between 1998 and 2010 as a solution to secure the land use rights of smallholder farmers and incentivize longer-term land-based investment and sustainable land use practices [16]. The program was implemented in the non-pastoral areas of the country including Amhara, Oromia, Tigray, and South Nations Nationalities and Peoples (SNNP) regional states [17]. This approach was praised for being low-cost, fast, large-scale, and participatory [4]. Despite documented benefits at the household level and to the environment [18–20], the FLLC had many limitations including a lack of geospatial information of registered parcels, a lack of unique parcel identification, and a paper-based land register that did not include any option to update or maintain the records in the case of a subsequent land transaction [16,20,21]. Though the FLLC was employed through a systematic land registration approach, there were significant numbers of unregistered parcels due to landholders' fear of confiscation, an increase in land taxation, or some other socially undesired norms [22]. Other research evidence from parts of Ethiopia suggests that customary property interests often go unregistered [23].

Since 2013, Ethiopia has continued investing in the second-level landholding certification (SLLC) program to address the limitations of the FLLC and maintain tenure security [20]. According to the Ministry of Agriculture (MoA) [24], between 2013 and the end of 2021, over 22 million rural parcels have been demarcated and mapped, of which close to 18 million parcels have been issued with SLLC. The SLLC program is still ongoing; it uses ortho-images as a base map for the cadastral surveying produced from aerial photography (25 cm) [25] and high-resolution satellite images (40 and 50 cm) [20]. The SLLC includes information on the parcel map (geographical location, size, shape, and land use), and textual information on the landholder(s) and their landholding rights and encumbrances [3,20,21]. It is notable that the FLLC and SLLC only adjudicate and register the existing landholding rights attached to the claimed and demarcated parcels [22]. In areas where the FLLC had been conducted, FLLC data were used as a legal source of information and verified by the kebele (village) land administration committee elected by the community and field para surveyors in the SLLC process [3,20]. After no objection obtained from the landholders or necessary correction made at the woreda (district) land offices, parcel-based SLLCs were prepared and approved by the pertinent authorities. Then, landholders were issued with the SLLC in hard copy and advised to report and formalize any subsequent land transactions. The formalization of land transactions helps to update the land register kept at the district land administration offices [26]. Alongside the SLLC, the Government of Ethiopia introduced complementary land market innovations to im-

prove access to the credit, land rental, and agricultural inputs markets that make the rural land administration system more sustainable [27].

To ensure the functionality and sustainability of the land registration system, rights holders are expected to fulfill the legal and administrative requirements of formalizing or registering subsequent land transactions [26]. How rights holders perceive and respond to the established process of formalizing subsequent land transactions after the first-time land certification is an important issue. For instance, the formalization of a new land transaction will have a cost associated with it in fees, stamp duty, and other transaction costs including travel to the service center [28]. After the successful implementation of the FLLC and ongoing SLLC programs over the past two decades, there is a growing and valid concern about the sustainability of the land registration system emanating from issues related to updating the land register to reflect continued land transactions in Ethiopia [28,29].

Recently, a few studies in Ethiopia have examined the functionality of updating the land register. For instance, with an exploratory and qualitative approach, Cochrane and Hadis [28] have studied the functionality of the land registration program in three woredas of the South Wollo administrative zones in northeast Ethiopia. They reported that updating the land certificates was functional in some instances but not in all and suggested investigating the causes of non-functionality and viable options addressing the causes thereof at scale. One of the reasons that was identified for non-functionality is woreda–kebele non-collaboration, with disconnected processes of updating at these levels. Another study was carried out by the Ethiopia Economic Association (EEA), who also reported that the volume of formal land transactions registered is still low [29]. On the other hand, Biraro et al. [3], using a qualitative and comparative approach, have studied good practices in updating the LIS in unconventional ways in nine case study countries, including Ethiopia, that apply similar data collection procedures for initial registration. Despite the existence of good practices in the cases of countries possessing the required infrastructure, they argue that the technical and financial sustainability of land registries in countries without the required infrastructure is uncertain and threatened by long registration procedures and a lack of data sharing among institutions [3]. In Rwanda, Ali and Deininger have also found that 5 years after completing a first-time land registration, 87% of rural transactions remain informal and the cost of registration is the main reason [30].

However, scholars in the field of land administration rarely study the formalization of subsequent land transactions through the lens of landholders' intentions. However, studying these intentions is important at least for two reasons. Firstly, the analysis of landholder's intentions helps to predict formalization rates. Secondly, it helps to improve our understanding of the factors that are responsible for the realization, or not, of these intentions. The focus of the current study is on the latter research stream. This information can help inform target interventions for different types of landholders facing various constraints in putting their intentions into action and ensure a functional and sustainable LIS.

This study demonstrates that the application of the theory of planned behavior (TPB) can be usefully employed to deepen our understanding of land rights holders' formalization of subsequent land transactions. Through the lens of TPB, among other issues, the authors considered and explored the influencing factors of rural land transaction formalization and their relationships, including attitudes, subjective norms, and perceived behavioral control as antecedents to the intention and actual behavior of land rights holders. In addition, the authors considered the role of background factors, such as institutional policies and regulations, societal customs and values, and personal characteristics.

## 2. Theoretical Base

The theory of planned behavior (TPB) evolved from the theory of reasoned action (TRA) [31]. TRA was developed in 1967 by Fischbein for the first time, in response to the repeated failure of traditional attitude measures to predict a given behavior [32]. The theory begins with the premise that the simplest and most efficient way to predict a given behavior was to ask a person whether he or she was or was not going to perform that

behavior [33]. Thus, according to the theory, performance and non-performance of a given behavior are primarily determined by the strength of a person's intention to perform (or not perform) that behavior, where the intention is defined as the subjective likelihood that one will perform (or try to perform) the behavior in question ($\rightarrow$ Attitude—Behavior consistency) [32]. During the early 1970s, the theory was revised and explained by Ajzen and Fishbein [33]. By 1980 the theory was used to study human behavior and develop appropriate interventions [34]. In 1988, the TPB was added to the existing model of reasoned action to address the inadequacies that Ajzen and Fishbein had identified through their research using the TRA [31,34].

The theory of planned behavior adapted the components of the TRA but added perceived behavioral control as an additional factor predicting both behavioral intentions and actual behavior [35]. Since the TPB was proposed by Ajzen in 1985 [36], it has attracted extensive interest and has been widely applied worldwide. To date, numerous studies have tried to improve the interpretation ability of the theory by extending variables or integrating other theories into the TPB model [37,38]. The TPB has received broad attention in social and behavioral sciences such as health, environment, business and management, educational research, and the political sciences [38]. Investigators continue to explore the intricacies of the structural model such as moderating the effects of perceived behavioral control and proposing additional factors to account for the complexity of human behavior [37].

At the core of the TPB is its concern with the predication of intentions [39]. According to the TPB, human behavior is guided by three kinds of considerations: beliefs about the consequence of the behavior, beliefs about the normative expectations of others, and beliefs about the presence of factors that may facilitate or hinder the performance of the behavior [31,34,38,39]. Put it in other ways, the beliefs can be influenced by behavioral beliefs, normative beliefs, and control beliefs. Behavioral beliefs produce favorable or unfavorable attitudes toward the behavior, whereas normative beliefs result in perceived social pressure or a subjective norm. In addition, control beliefs give rise to perceived behavioral control or self-efficacy [34,39,40]. Whether intentions predict a behavior depends in part on factors beyond the individual's control, meaning the strength of the intention to behavior relation is moderated by actual control over the behavior [39]. The effects of attitude towards the behavior, and subjective norm on the intention, are moderated by the perception of behavioral control [38]. As a rule, the more favorable the attitudes and subjective norms, and the higher the perceived behavioral control, the greater the person's intentions should be to perform the behavior in question [34,38].

According to the TPB, attitudes towards the behavior are defined as the individual's positive and negative feelings about performing a behavior [41]. It is determined through an assessment of one's beliefs regarding the consequences [34]. Likewise, subjective norms are defined as an individual's perception of whether people important to the individual think the behavior should be performed [41,42]. The contribution of the opinion of any given referent is weighted by the motivation that an individual must comply with the wishes of that referent [42]. In summary, by changing these three 'predicators', i.e., attitudes, subjective norms, and perceived behavioral control, we can increase the chance that the person will intend to do the desired action and thus increase the chance of the person doing it or the actual behavior [31,34,38].

Regarding the land registration system, Abubakari, Richter, and Zevenbergen [11], in their recent work, highlighted three general underlying assumptions for the conventional land registration approach and show what such assumptions imply for the emerging fit-for-purpose approaches. According to these authors, the three underlying assumptions include (a) desirability, (b), registrability, and (c) accessibility [11]. The arguments outlined by Abubakari, Richter, and Zevenbergen are much closer to what is stated in the theory of planned behavior. Likewise, Deininger and Feder also underlined the need to understand the context while introducing a land registration system in each jurisdiction [4]. Overall, individuals are located within a social 'culture' that influences the development of values, beliefs, attitudes, and behavior [34]. Socio-cultural factors play an influential role in shaping

an individual's early life (and later life) experiences and general beliefs about the world [43]. Hence, access to the land registration system is embedded in socio-cultural practices of land allocation, and practices of landholding, as well as the practice of land registration [4,11,44].

Similarly, Biraro et al. [3] also identified comprehensive criteria in unconventional contexts of updating land information. Bennett et al. [12] also outline aspects of land administration maintenance including the level of change, method of change, components, and options to change. Zevenbergen in his early work on the system thinking approach to land registration, outlined aspects and conditions of land registration systems including the initial establishment and updating [2]. Magis and Zevenbergen [45] have also highlighted long-term value creation mechanisms of land registration for sustainable land administration system design. This major scholarly literature covers what should be considered in establishing, maintaining, upgrading, and updating the function of a land registration information system for sustainability.

Based on this brief theoretical analysis, the authors posit the following theoretical assumption. As the desired outcome of a functional and sustainable land registration system, land rights holders' formalization (registration) of subsequent land transactions is an outcome of their intentions. The authors preferred formalization over registration for subsequent land transactions to minimize confusion of concepts with first-time land registration. According to Ethiopia's land administration legislation, the formalization of subsequent land transactions includes 18 business cases related to inheritances, gifts, rentals, divorce, parcel mutation or merger, collateral, and encumbrances [26,27]. Land rights may be transferred partially or fully for a certain period or perpetually. The formalization of these subsequent land transactions is dependent on the perceived behavioral belief or attitudes of the land rights holders toward the legal consequences of land registration. That is, the decision to formalize a transaction is an output of the value placed by individual landholders or groups of people on the benefits of formally registering one's land use or holding rights [45]. These benefits include secure tenure, control and/or use, and transfer rights to others [27].

In terms of the enforcement of rights, land registration is a dynamic and complex system between individual titling decisions and social choice [10]. The registration system secures all registered parcels [2,3] and needs to be designed to facilitate updates to the land register following transactions [45], which implies a cost. The formalization of subsequent land transactions by the land rights holders is also influenced by beliefs about the normative expectation of others (normative beliefs) or referents such as a spouse, relatives, and extension service providers. Moreover, land rights holders' beliefs about factors that may enable, or hinder land transactions registration (control beliefs) also affect their intentions to formalize subsequent land transactions. The control beliefs can include the accessibility, affordability, simplicity of procedures, speed/time, and incentives(benefits) of formalization, among others [3]. Overcoming such control beliefs would determine the level of self-efficacy [34] that in turn influences the intentions and actions of formalization of the land rights holders.

Landholders' intentions to formalize land transactions and their actual behavior are assumed to be of particular interest to policymakers mainly for two reasons. Firstly, knowing the intention to formalize would help to estimate the rate of transactions for planning and resource allocation. Secondly, it would also help to improve understanding about what factors influence land rights holders' intentions to formalize subsequent land transactions. Hence, appropriate interventions to encourage land rights holders to formalize subsequent land transactions could be designed to strengthen the land registration system. Therefore, facilitating landholders' attitudes of formalizing land transactions, will help them to overcome social norms and control beliefs and will likely increase intentions and actual practice that ensures land registration systems remain up to date and sustainable. Hence, the following hypotheses were constructed and tested.

**Hypothesis 1 (H1).** *Attitudes of land rights holders towards the value of land registration have a significant influence on the intentions of land rights holders to formalize subsequent land transactions.*

**Hypothesis 2 (H2).** *Attitudes have a significant influence on the overall subjective norms that influence land rights holders' formalizing subsequent land transactions.*

**Hypothesis 3 (H3).** *Attitudes have a significant influence on the perceived behavioral control towards land rights holders' formalization of subsequent land transactions.*

**Hypothesis 4 (H4).** *Subjective norms have a significant influence on land rights holders' intentions to formalize subsequent land transactions.*

**Hypothesis 5 (H5).** *Perceived behavioral control has a significant influence on land rights holders' intentions to formalize subsequent land transactions.*

**Hypothesis 6 (H6).** *Perceived behavioral control has a significant influence on the formalization of subsequent land transactions by the holders of land rights.*

**Hypothesis 7 (H7).** *Intentions to formalize subsequent land transactions have a significant influence on land rights holders' formalizing of subsequent land transactions.*

The theory and hypotheses are presented in Figure 1.

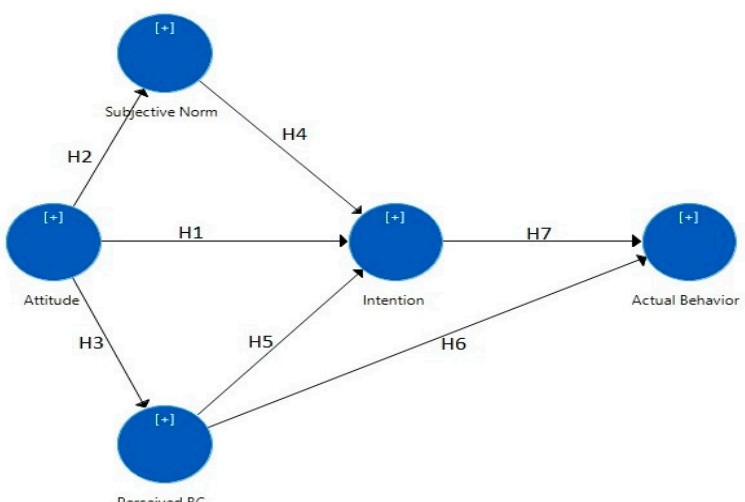

**Figure 1.** Research model with latent variables and hypotheses adapted from Icek Ajzen theory of planned behavior and applied to the formalization of subsequent land transactions.

### 3. Study Site and Methods

This section presents explanations about the study area contexts and the methods employed for data collection including sampling methods, sample size, data collection instruments, and materials used to generate the required data for this study. This section also describes the methods of analysis and means to triangulate the results of the quantitative and qualitative data.

#### 3.1. Study Site

The Basona Worena woreda is one of the rural woredas selected for the household survey of this study in the Amhara regional state. It is situated surrounding the Debre Birhan town, the North Shewa administrative zone's capital, and is located 130 km from northeast Addis Ababa, Figure 2. Debre Birhan is a historic town located at the very heart of the medieval age history of the country. It is one of the fastest-growing zonal administration towns of the Amhara regional state recently. Debre Birhan is expanding in all directions to its surrounding Basona Worena woreda. It is becoming an industrial investment hub for hundreds of mega, medium, and small-scale industries. This fuels the expansion of new residential neighborhoods to labor households and exacerbates horizontal expansion. As a

result, Debre Birhan has expanded its urban form and size threefold into the neighboring Basona Worena woreda over the past decade. This fast urban growth leads to a high agricultural land conversion into the built environment. Therefore, the competition of land resources between urban development and agricultural practices is considerable and the Basona Worena woreda land resource is the podium to this arena. This implies that there would be a high potential for land transactions in the study woreda.

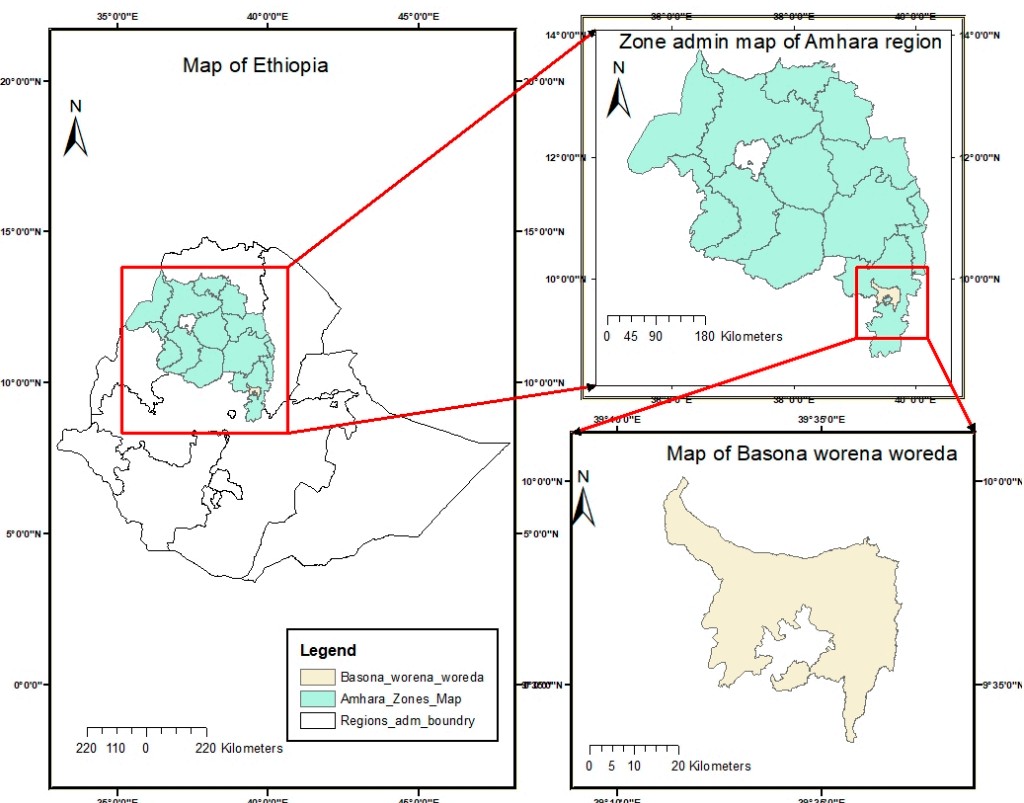

**Figure 2.** Study Site Map. Map organized by the authors based on the 2018 updated shape file/map of Ethiopia by Central Statistical Agency under the auspices of the Ministry of Finance.

The Basona Worena woreda is composed of 30 rural kebeles with a population of around 135,000 persons and 27,000 households, according to the information received from the woreda land administration office. It covers around 140,000 ha and close to 250,000 land parcels. The Basona Worena woreda represents highland agroecology where mixed farming is the major source of livelihood. It has an altitude range of 1300–3400 m above sea level [46]. According to the woreda land office, the woreda has 42,828 hectares of arable land with 159 hectares of irrigable land, and 6828 hectares of forest plantations. Crops grown in the Basona Worena woreda include teff, barley, wheat, faba bean, field pea, sorghum, lentil, chickpea, onion, potato, temperate fruits, and oil crops [46].

The Second Level Land Certification (SLLC) issuance process was conducted between 2016 and 2019. The SLLC was supported by the UK-funded Land Investment for Transformation (LIFT) program and resulted in the identification of close to 250,000 parcels, where 99.6% of the landholders received the SLLC. The installation of a national rural land administration information system (NRLAIS), a digital land register and data migration system, has been initiated since September 2019 and has been operational since 2020. According to the key informants, the Basona Worena woreda also undertook the FLLC program between 2005 and 2007. Between 2008 and 2010, the woreda land administration office converted the manual land register into a digital land register called the Information System for Land Administration (ISLA). ISLA is a pioneer computerized land registration software solution designed to operate in the woreda land offices in the Amhara region. ISLA supports the registration of parcels, landholders, and their rights for the transfer of parcels and attached

rights including division of parcels, leases, and easements, and it facilitates statistical reporting [47]. According to the key informants, the ISLA data have been instrumental for the implementation of the SLLC program in the Basona Worena woreda.

The Basona Worena woreda has a land office in Debre Birhan, the seat of both the zonal and the woreda administrations. The woreda land office has three business teams; the land administration work process team is one of the three teams and is staffed with seven experts. There are also 30 land administration experts—one expert in each rural kebele land office. Besides this, there are elected and voluntary land administration committees that are responsible for the facilitation of the land registration and certification process and land disputes resolution, among others. The kebele land offices are the windows for landholders to lodge their applications or land services, including registration of subsequent land transactions. Accordingly, survey data was collected from land rights holders of seven different rural kebeles (villages) of the Basona Worena woreda (district).

### 3.2. Methods for Data Collection and Analysis

### 3.2.1. Data Types and Collection Methods

The study employs primary and secondary data. The primary data were collected from smallholder households through a cross-sectional survey and key informant interviews (KIIs) with woreda and kebele land administration experts. Key informant interviews were employed to collect information on standards, procedures, requirements, trends, types, and magnitudes of subsequent land transaction registrations. The KIIs were also intended to learn about the understanding, knowledge, skills, attitudes, experiences and perception of respondents on subsequent land transaction handling. In addition, the secondary data were collected through a review of the scientific literature, program documents, operational manuals, and data mining from the woreda digital land register, called the national rural land administration information system (NRLAIS).

A pilot survey questionnaire was developed for the study based on the guideline for the TPB questionnaire [48]. According to the guideline, before any work begins, the behavior of interest must be clearly defined in terms of its target, action, context, and time elements [49]. However, to elicit salient beliefs initially, the authors first visited the woreda land administration office of the study area in February 2021, where landholders formalize subsequent land transactions that they have made after first-time land certification. This first stage was carried out individually in a free-response format related to readily accessible behavioral outcomes, normative referents, and control factors. Next, five to six formative items were formulated to develop the pilot survey questionnaire for direct measures. Seven-point bipolar adjective scales were employed. The respondents were asked to state their opinions using a seven-point Likert scale, from strongly disagree (1) to strongly agree (7), for behavioral beliefs, referents, and behavioral control measurement-related items. The formulation of the pilot questionnaire was aimed to assess each of the theory's major constructs: attitude, subjective norms, perceived behavioral control, intention, and the actual behavior [48,49] of landholders towards reporting and formalizing subsequent land transactions. The pilot questionnaire also included measures of background factors and other variables, including demographic characteristics and other socioeconomic data.

Using the pilot questionnaire, 28 landholders, 12 of whom were female, were interviewed in the second week of February 2021. Most of the landholders had visited the woreda land administration office to follow up on their earlier application of updating land transactions including inheritance, donation, lost certificate replacement, and complaints related to land-related conflicts. The results of the pilot questionnaire allowed the authors to evaluate the validity and consistency of each item and the utility of the background measures, and other socioeconomic indicator items. Based on these inputs, necessary adjustments were made and the standard questionnaires to be used in the main study were produced.

Tablet-based data collection was employed for the actual survey from April to May 2021. After successful survey questionnaire pilots and adjustment, the questionnaire

was programmed into a cloud-based Kobo Toolbox server. An Open Data Kit (ODK) data collector application was installed and configured on Android tablets to connect, download, fill, and upload the survey questionnaire to and from the cloud server. Before the actual survey, training was provided to eight data collectors and two supervisors on the use of the tablet-based data collection, management of the tablet, and hardware to limit complications in the field. Moreover, orientation on the meaning of each item, protocols for the tablet-based data collection, procedures on addressing data inconstancies or misreporting, data transmission, validation protocols, contingency plans, and steps, if needed, to revert to a paper-based version were part of the training. The tablet-based data collection was a systematic approach that reduced data entry error as the questionnaires were encoded with prepopulated entries based on prior information. The tablet-based data collection also improved the quality of the data and saved time per interviewee. The average time taken to complete a household survey was 40 min.

### 3.2.2. Data Analysis Methods

A partial least square structural equation model (PLS-SEM) and descriptive statistics were used for the quantitative survey data analysis. Qualitative data collected through key informant interviews were first organized thematically. Then, the analysis focused on context-specific meanings and explanations for reported practices, and perceptions of land experts on processing subsequent land transactions. Next, a deductive approach that triangulates and explains results from the quantitative survey data was applied. A summary of the descriptive and inferential statistics was also employed to analyze the results of the survey data related to socioeconomic and demographic characteristics of the respondents.

PLC-SEM uses proxies of interest, which are weighted as composites of indicator variables for a particular construct; in most cases, this is considered a novel second-generation multivariate statistical technique [50]. Smart-PLS software version 3.0 was used to process the data analysis related to the coefficient of interaction terms [51]. PLS-SEM includes confirmatory factor analysis, path analysis, and partial least squares to impute relationships between latent variables [52]. In PLS-SEM, the model has two parts namely, the outer or measurement model and the inner or structural model. In the measurement model the authors' assessed and evaluated whether the constructs of the latent variables were valid and reliably constructed. The validity and reliability of the measurement model of the research were tested using the factor loading analysis of each latent variable. Whereas in the structural model, the authors tested the path hypothesized in the research framework.

- Validity

Validity tests attempt to determine how accurately or well an instrument measures a particular concept and it is designed to measure the underlying construct [50]. This shows the extent to which the items used to measure can calculate the idea that they are meant to quantify [53]. The measurement model shows the subjective independence of every indicator on its latent variable using factor loading or cross-loading criteria [52]. Through factor loadings, the convergent and divergent validity of each item was evaluated against the underlying latent variable construct. The individual item reliability was evaluated by examining the loading and cross-loading of indicators in their respective constructs. Moreover, Composite Reliability (CR) and Average Variance Extracted (AVE) analyses were used to test the convergent reliability of the constructs [53], whereas a divergent validity test was carried out using the Fornell–Larcker analysis and Heterotrait–Monotrait Ratio (HTMT). The divergent or discriminant validity examines whether that constructs' items correlate with the items of other constructs [54].

- Reliability

A reliability test tries to find the stability and consistency of the measuring instrument. Composite reliability and Cronbach's alpha were used to test the reliability of each construct [50]. The composite reliability and Cronbach's alpha examine whether the items

are measuring what they are supposed to measure. Put differently, the Cronbach Alpha analysis examines the consistency of data. According to Fornell and Larcker's criteria, a reliability score of Cronbach Alpha 0.6 is considered minimally acceptable, with 0.70 preferred (50% of the explained variance) [54]. This theory also recommends that an indicator loading with a value less than 0.4 should be removed from the model [50].

In the second part of the research model, a structural model was used to test the structural relationships between the hypotheses. Using the bootstrap resampling technique (5000 resamples), the path coefficient was tested to examine the significance of the hypothesis. A t-value > 1.96 is considered significant at $p < 0.05$, and a t-value > 2.58 is significant at $p < 0.01$ [50]. Most researchers also use $p$ values to assess significance levels. When assuming a significance level of 5%, the $p$-value must be smaller than 0.05 to conclude that the relationship under consideration is significant at a 5% level [50].

The structural model was also assessed based on the $R^2$, $Q^2$, and significance of paths. The $R^2$ indicates the proportion of the variance for the endogenous variable explained by the exogenous variable [50,55]. The validity of the model was determined by the strength of each structural path, determined by the $R^2$ value for the dependent variable; the value for $R^2$ ranges from 0 to 1, with higher levels indicating higher levels of predicting accuracy [50,52,55]. According to Hair et al. [53], $R^2$ values of 0.75, 0.50, and 0.25 can be described as substantial, moderate, and weak, respectively [53] while Chin articulated $R^2$ values of 0.67, 0.33, and 0.19, considered as substantial, moderate, and weak, and evaluated subsequently. Hence, the predictive capability is established. Moreover, $Q^2$ establishes the predictive relevance of the endogenous construct. A $Q^2$ above 0 shows that the model has a predictive relevance [50].

### 3.3. Sampling Method

To determine the sample size required for a study that uses a structural equation model (SEM), the authors applied the Soper [56] online free statistic calculator, which calculates priori sample sizes for structural equation models. This sampling calculator considers the number of observed and latent variables in the model, the anticipated effect size, the desired probability, and the statistical power levels, presented in Table 1. The model of this study contains 20 observed variables and six construct latent variables. The model considers the anticipated medium effect size of 0.3, the desired probability level of 0.05, and the desired statistical power level of variables of 0.8. Hence, the minimum initial sample size to detect the effect was determined to be 150, the minimum sample size for the model structure was 100, and the recommended minimum sample size was 150.

**Table 1.** Sampling parameters and values.

| Parameter | Values |
| --- | --- |
| Anticipated effect size: medium | 0.3 |
| Number of latent variables | 6 |
| Probability level | 0.05 |
| Desired statistical power level | 0.8 |
| Number of observed variables | 20 |
| Minimum initial sample size to detect an effect size | 150 |
| The minimum sample size for the model structure | 100 |
| Recommended minimum sample size | 150 |

A systematic random sampling method was used to identify the sample kebeles of the study woreda. There are 30 kebeles in the Basona Worena woreda, of which seven kebeles were randomly selected including Bakelo, Billila, Birbisa, Debele, Goshebado, Mehal Amba, and Wushawishign. As a result of these sample size requirements, a careful review of the existing land register in the woreda land office was carried out. The total landholder

population in these sample kebeles was 11,149. To randomly select a proportional sample of respondents, the authors used the landholding identification numbers in the ISLA computerized land register and ran the sampling model in Microsoft excel. Hence, 320 respondents were identified randomly from all sample kebeles including reserve respondents. However, data were collected from only 206 respondents, which is above the recommended minimum sample size of 150 respondents for the research model.

## 4. Results

### 4.1. Characteristics of Respondents

Discussing the demographic features and socioeconomic conditions of respondents would have a significant role in understanding whether such a background affects the intention and decision on the formalization of subsequent land transactions. According to the survey result, the sample respondents included 148 (72%) male and 58 (28%) female smallholder household heads, Figure 3. This indicates that most landholder households are male-headed in the study kebeles. The age of the respondent's ranges between 20 and 93 with a mean of 49.31 years. This may affect the socioeconomic aspects of the households with far reaching implications on access to land rights, registration of the subsequent land transaction, and land management practices in the study area.

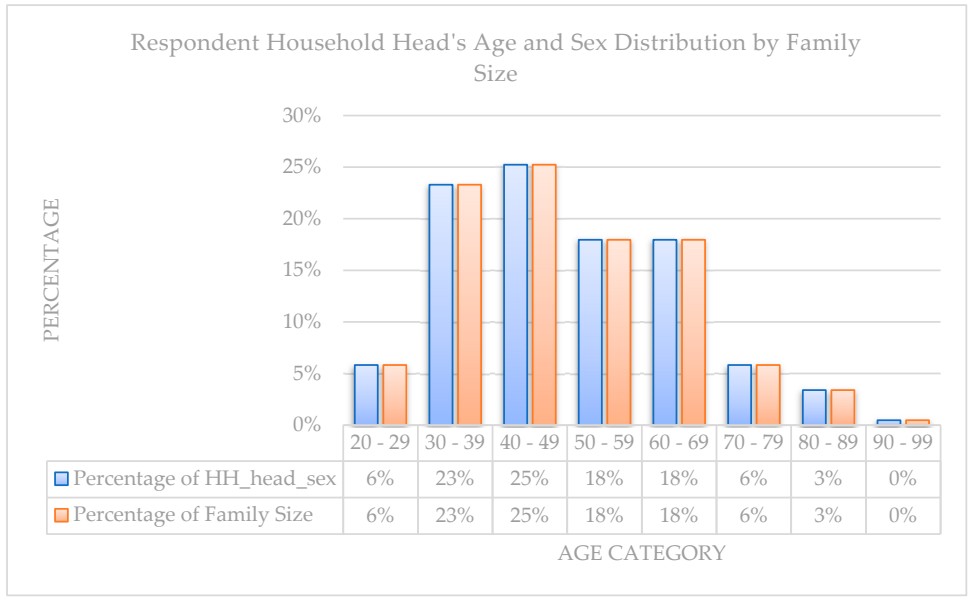

**Figure 3.** Respondent household's age and sex distribution with family size. Source: authors' compilation from household survey.

According to the Ethiopia Statistical Service (ESS), ages between 15 and 65 are generally considered as the working ages of the population. The result indicates that the majority of the respondent landholders (about 90%) are in their working ages. In addition, the sample respondents' family size ranges between 1 and 10, with a mean family size of 4.31 and a total of 888 persons, of which 47.75% were females, including household heads. Family labor is one of the most important assets of the respondent households. Hence, the availability of sufficient labor within respondent households would help them to undertake better land use management practices and increase land productivity.

Regarding respondents' educational status, about a third of them were illiterate, while 9.17% can read only, another 34.47% can read and write, whereas the remaining quarter (24.27%) completed 4th-grade and above, Table 2. The survey result also revealed that two thirds of all illiterate respondents were men. This shows that a third of the respondents require some sort of literacy aid from others who can read them some documents related to their landholding rights and registration process. In addition, about 78% of all men (148) and 38% of all women (58 respondents) were married landholders, Table 2. This implies

that there are more married women who enjoy access to land, jointly with their male counterparts, than unmarried women (27) who have access to land use right independently.

**Table 2.** Respondent's Marital and Educational Status by Sex. Source: Authors' Compilation from the household survey.

| | Educational Status | | | | | | | |
|---|---|---|---|---|---|---|---|---|
| | Female | | | | | | | |
| Marital Status | >12 Grades Complete | Grade 10–12 Complete | Grade 8 Complete | Grade 4 Complete | Read and Write | Read-Only | Illiterate | Female Total |
| Divorcee | 0 | 0 | 0 | 1 | 0 | 0 | 0 | 1 |
| Married | 0 | 2 | 1 | 1 | 9 | 1 | 8 | 22 |
| Unmarried | 0 | 1 | 10 | 3 | 7 | 0 | 6 | 27 |
| Widower(ed) | 0 | 0 | 0 | 0 | 0 | 0 | 8 | 8 |
| Sub Total | 0 | 3 | 11 | 5 | 16 | 1 | 22 | 58 (28%) |
| | Male | | | | | | | Male Total |
| Divorcee | 0 | 0 | 0 | 0 | 0 | 0 | 2 | 2 |
| Married | 0 | 4 | 4 | 9 | 47 | 17 | 34 | 115 |
| Unmarried | 1 | 5 | 3 | 5 | 6 | 2 | 6 | 28 |
| Widower(ed) | 0 | 0 | 0 | 0 | 2 | 0 | 1 | 3 |
| Sub Total | 1 | 9 | 7 | 14 | 55 | 19 | 43 | 148 (72%) |
| Grand Total | 1 (0.5%) | 12 (5.8%) | 18 (8.7%) | 19 (9.2%) | 71 (34.5%) | 20 (9.7%) | 65 (31.5%) | 206 (100%) |

### 4.2. Landholding Assets of Respondents

Land is the primary source of livelihoods for all respondent households. Despite difference in the means of acquisitions, the size of a landholding reflects an important productive asset and socioeconomic status of respondents within their own community. This socioeconomic status would affect respondent's intentions and decisions in formalizing subsequent land transactions. In terms of landholding assets, Table 3 shows that the total area of landholding possessed by the respondents ranges between 0.06 and 7.8 hectares with a mean area of 1.83 hectares in the study area. According to the survey result, the number of parcels per landholding ranges between 1 and 6 with a mean of 4.15 parcels per respondent, which is a bit above the national average, Table 3. This indicates a high fragmentation of landholding in the study area, which may have implications on the cost of subsequent land transactions registration in the long-run.

**Table 3.** Summary and descriptive statistics of selected variables.

| Variable | Observations | Mean | Std. Dev. | Min | Max |
|---|---|---|---|---|---|
| Family size | 206 | 4.31 | 2.32 | 1 | 10 |
| Household head sex (M = 1, F = 0) | 206 | 0.72 | 0.45 | 0 | 1 |
| Household head age | 206 | 49.31 | 14.36 | 20 | 93 |
| Total number of parcel | 206 | 4.15 | 1.63 | 1 | 6 |
| The total area of landholding | 206 | 1.83 | 1.17 | 0.06 | 7.8 |
| SLLC received | 206 | 1.04 | 0.39 | 1 | 6 |
| Willingness to pay for SLLC | 206 | 174.85 | 93.23 | 0 | 300 |

Moreover, a sizable number of parcels of landholdings were found below the minimum landholding size determined by the Amhara regional state revised rural land administration and use proclamation No. 252/2017 [57] and regulation No. 159/2018 of the same [58], Table 3. The regulation, Article 5.1, determined that the minimum parcel size that rural landholders are entitled to obtain in the regional state may not be below 0.25 hectares for rain feed, 0.06 hectares for irrigable parcels, and 0.02 hectares should it be used for the construction of a dwelling. Article 5.2 of the same regulation states that "in any conditions and settings, no parcel (s) of a holding shall subdivide below the limit indicated under Article 5.1." This legal precondition, while encouraging the defragmentation of parcel(s) of a holding, may impede the transfer and registration of some transactions and leave it under informal conditions.



Despite having a similar average family size and number of parcels per landholding to that of the national average, the results show low levels of per capita landholding size (0.42 ha) in the study area. This characterizes a typical land scarcity in the study area that reflects the scarcity in the highland regions of the country. Since the parcels are below the minimum size, this may affect the land acquisition opportunities among heirs, such as through inheritance and gift, in the study area. Further, this could affect the registration of such land transactions after the first-time land certification. The fragmentation may also increase the cost of sustainable land use management practices of the households in the study area.

The survey result also revealed that 26.7% of all respondent households had been engaged in a subsequent land transaction since 2019. The year 2019 hallmarks the completion of the SLLCs issuance in the study area. These subsequent land transactions consist of inheritance with a will (4%), a gift (2.9%), rent (9.7), and access to credit (9.2%). The survey result also notably revealed that all inheritance and access to credit transactions had been registered in the land register. Whereas only 16.7% of all gifts and 35% of all land rentals were registered or formalized in the woreda land register. This shows that the majority of land transactions made through gifts and rentals were not found formalized, which affects the up-to-datedness, accuracy, and completeness of information in the land register of the woreda. Respondents also revealed that the reasons for lack of registering all rentals and gift transactions was because a land transaction among family members or relatives needed no registration since customs warranty the protection and enforcement of contracts. This information and trend is quite similar with the data generated from the woreda digital land register, explained in Section 4.4 below.

*4.3. Distance to the Woreda Land Office, Cost of Travel and Registration*

Distance to the land registration service center, in this case the woreda land administration office, was assessed with the aim to factor its effect on respondent's intentions to formalize subsequent land transactions. According to the survey result, 98% of landholders have received an SLLC as a first-time certification for at least one of their parcels of landholding by 2019. The average distance to travel to the woreda land office ranges from 9 to 28 kilometers from the respondents' respective resident kebeles, Table 4. This allowed us to compare the likelihood of intentions to formalize subsequent land transactions. The survey result also revealed that the average cost of travel to the woreda land office per round trip, including purchasing a normal lunch, ranges from 190 to 383 Ethiopian Birr (ETB), should they be using a public transport. Though all the sample kebeles have access to public transport, the type of road varies between asphalt and all-weather gravel roads, as do the tariffs of travel cost. It was found that land rights holders closer to the woreda land office with access to public transport with an asphalt road have a much higher intention and practice of formalizing subsequent land transactions.

**Table 4.** Average distance to woreda land administration office, cost of travel/round trip, and willingness to pay for the registration of a subsequent land transaction. Source: Authors' compilation from household survey.

| Kebele Name (ID) | Average Distance to Woreda Land Office (KM) | Average Traveling Cost (ETB) | Average Willing to Pay for the Registration of a Subsequent Land Transaction | Total Cost Should the Registration of a Subsequent Land Transaction Be Complete in a Single Round Trip |
|---|---|---|---|---|
| Bakelo (4) | 14 | 190 | 203 | 393 |
| Debele (9) | 24 | 282 | 155 | 437 |
| Birbisa (12) | 9 | 158 | 157 | 315 |
| Goshebado (14) | 19 | 362 | 209 | 571 |
| Wushawishign (20) | 28 | 383 | 146 | 529 |
| Mehal Amba (27) | 15 | 217 | 153 | 370 |
| Billila (30) | 17 | 311 | 172 | 483 |
| Grand Total Average | 18 | 272 | 171 | 443 |

Another interesting element considered related to cost was the registration and service charges. Land administration service fees have a direct impact on the perceived behavior or willingness to register subsequent land transactions and keep the land register up-to-date [3,12]. The survey result also shows that landholders are willing to pay up to 300 Ethiopian Birr (ETB) per SLLC with a mean value of 175 ETB, Table 3. The average willingness to pay, per kebele, ranges from 153 ETB to 209 ETB, Table 4. In addition, the survey result shows that the average total cost (cost of travel and registration fee) differs from kebele to kebele. It was found that, should the registration of a subsequent land transaction be complete in a single round trip, per kebele, the cost ranges from 315 ETB (respondents from Birbisa—the nearest kebele with a gravel road) to 571 ETB (respondents from Goshebado—the second most remote kebele with a gravel road). Likewise, the average total cost of travel and a subsequent land transaction registration for kebeles having access to asphalt public transport ranges from 393 ETB (respondents from Bakelo—the nearest kebele) to 483 ETB (respondents from Billila—the second most remote kebele).

According to the key informants, landholders are asked to pay 50 ETB for a transaction that needs updating, for those landholders who had received a first-time landholding certificate. Whereas, if the applicant is a new landholder who did not receive a landholding certificate in the first-time registration and certification, she/he is asked to pay 150 ETB for the SLLC, plus a 100 ETB service charge. Interestingly, despite this variation, the subsequent land transactions registration fee structures of the woreda are much more consistent with and within the range of what is found in the survey results of this study. The result revealed that there are good attitudes towards land registration values and willingness to pay the indicated registration and service fee, including the cost of travel. With proper interventions, this, in turn, indicates a good potential for a self-financing land registration system.

### 4.4. Results from Key Informant Interviews

Results from the key informant interviews (KII) also revealed that the woreda land office receives applications from clients on Mondays and Fridays. The remaining business days are dedicated to handling and processing the applications internally. According to the KII, the reception days and hours are well communicated to, and known by, land rights holders. It was mentioned that there are close to twenty buisness transaction types recognized by the land office. To faciliate these transactions, 15 types of standard application forms and copies of all of these are distributed to the kebele land offices, where applicants initiate their cases. It was claimed that all kebele land administration experts and the kebele land administration and use committees (KLAUC) have received proper training and have adequate knowledge to handle the different business case applications.

According to the report generated from the digital land register of the woreda land office, only about 3391 land transaction applications were processed and completed in 2021, Figure 4. This includes inheretance without a will (868 cases), ex-officio (882 cases), mortgage (604 cases), simple correction of tenure documents (458 cases), gift (285 cases), and expropriation and compensation (168 cases). In addition, completed transactions with a small number of cases include land-to-land exchange (66), divorce (19), replacement of land certificates (17), land rentals (15), land consolidation (5), and inheritance with a will (4 cases). This shows that over about half of all transactions were inheritance without a will and ex-officio cases, while inheritenace by will and parcel consolidations represented the least number of transaction of all registered transactions.

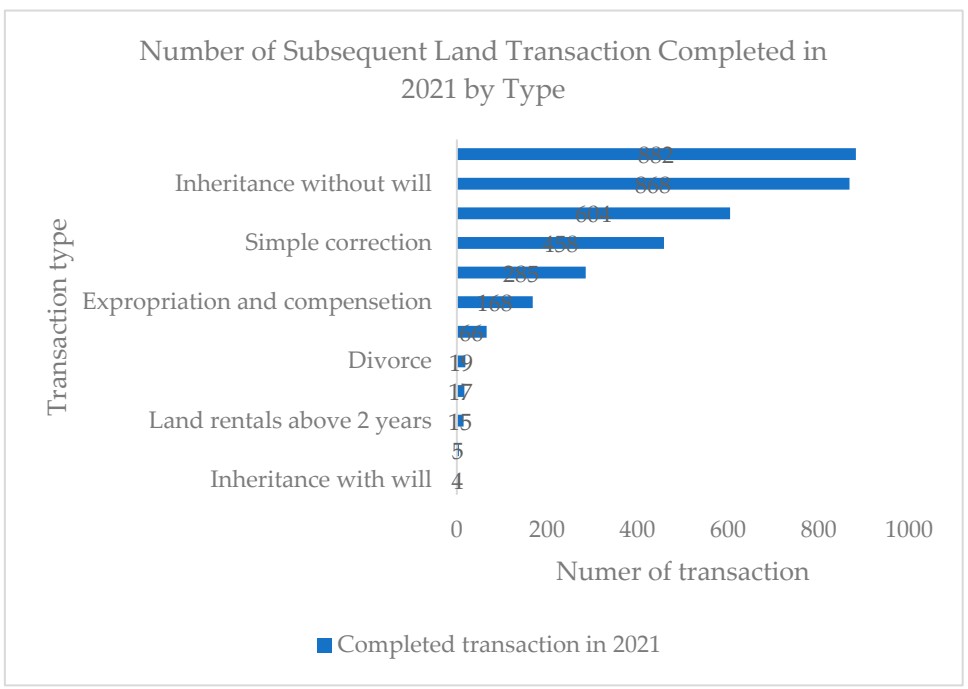

**Figure 4.** Numbers and types of subsequent land transactions completed in the year 2021. Source: Basona Werena woreda land office digital land register, called NRLAIS.

Key informants also revealed that land rights holders are becoming more interested in formalizing subsequent land transactions now than ever before. This is due to the continued awarness-raising compaigns made to landholders, KLAUC, and kebele land experts. However, they have also admited that, in some cases, more and continous awareness compaigns are needed, particularly on the benefits of registering subsequent land transactions and the risks of informal transactions. In addition, the introduction of new benefits of registration, such as SLLC-linked individual loans from micro finance institutions (MFIs) to landholders, incentivize them to have strong intentions to formalize subsequent land transactions.

However, key informants also revealed that sizable portions of subsequent land transactions remain unregistered and fall under an informal transaction regime. Some of the main reasons outlined for non-registered transactions include incomplete information of transacted parcels, missing supporting documents such as marriage certificates, transactions made of a parcel with no SLLC, fear of land taxation increases, the limit determined on minimium holding size, long administrative procedures, some disabled land rights holders being unable to come to the woreda office, limited awareness, no perceived need to register due to no perceived risk, and remoteness of the woreda office to some of the kebeles.

### 4.5. Validity and Reliability of the Analysis of the Structural Measurement Model

As per the rule outlined in Section 3.2.2, this study found and removed two indicator items with less than or equal to 0.4 outer loadings; these included SN1—*respondents do not believe formalizing subsequent land transactions within six months is important*) and PBC3— *respondents always forget to report and formalize subsequent land transactions that were made within six months*. These two items failed to reflect the underlying assumptions or constructs about subjective norms and perceived behavioral control, respectively. Otherwise the result revealed that all construct items have a moderate to strong content validity, Table 5.

**Table 5.** Indicator item cross-loading. Attitudes (AT), Subjective Norms (SN), Perceived Behavioral Control (PBC), Intentions to Formalize (INTF), and Actual Behavior/formalization (ACB).

| Variable | Code | AT | SN | PBC | INTF | ACB |
|---|---|---|---|---|---|---|
| Attitudes | AT1 | 0.879 | 0.578 | 0.281 | 0.652 | 0.790 |
| | AT2 | 0.924 | 0.627 | 0.262 | 0.710 | 0.804 |
| | AT3 | 0.869 | 0.545 | 0.236 | 0.621 | 0.658 |
| | AT4 | 0.886 | 0.654 | 0.245 | 0.665 | 0.745 |
| | AT5 | 0.809 | 0.582 | 0.114 | 0.608 | 0.639 |
| Subjective Norms | SN2 | 0.365 | 0.409 | 0.476 | 0.081 | 0.769 |
| | SN3 | 0.481 | 0.472 | 0.442 | 0.038 | 0.712 |
| | SN4 | 0.202 | 0.290 | 0.190 | 0.208 | 0.489 |
| | SN5 | 0.546 | 0.657 | 0.577 | 0.117 | 0.796 |
| Perceived Behavioral Control | PBC1 | −0.101 | −0.001 | 0.062 | −0.296 | −0.036 |
| | PBC2 | 0.206 | 0.206 | 0.143 | 0.741 | 0.086 |
| | PBC4 | −0.249 | −0.184 | −0.177 | −0.744 | −0.137 |
| | PBC5 | 0.074 | 0.073 | −0.037 | 0.096 | −0.072 |
| Intention to formalize | INTF1 | 0.628 | 0.496 | 0.284 | 0.855 | 0.663 |
| | INTF2 | 0.610 | 0.484 | 0.140 | 0.842 | 0.552 |
| | INTF3 | 0.632 | 0.617 | 0.067 | 0.811 | 0.597 |
| Actual Behavior/formalization | ACB1 | 0.781 | 0.538 | 0.268 | 0.635 | 0.925 |
| | ACB2 | 0.778 | 0.583 | 0.318 | 0.714 | 0.942 |

In the measurement model, Table 6, the study used Cronbach's alpha, composite reliability (CR), and AVE to test the reliability of the constructs. According to the rule, Cronbach's alpha and CR values of 0.700 and higher are preferred, with an AVE above 0.500. As can be seen from Table 6, except for the perceived behavioral control (PBC = 0.014), the CRs of all the other variables were higher than the recommended value of 0.700, ranging from 0.791 to 0.942. Similarly, the AVE of the PBC (0.300) was less than the recommended 0.500 and the AVE of the SN (0.493) was nearly 0.500. However, the AVE of AT (0.764), INTF (0.700), and ACB (0.872) were well above the recommended value of 0.500. Hence, the convergent validity was acceptable.

**Table 6.** Reliability and Validity. The "Rho_A" coefficient also considered to check the reliability of the latent variable construct scores, as defined in Dijkstra and Henseler (2015a).

| Variable Code | Cronbach's Alpha | Rho_A | Composite Reliability (CR) | Average Variance Extracted (AVE) |
|---|---|---|---|---|
| AT | 0.922 | 0.926 | 0.942 | 0.764 |
| SN | 0.659 | 0.711 | 0.791 | 0.493 |
| PBC | −0.058 | 0.250 | 0.014 | 0.300 |
| INTF | 0.785 | 0.786 | 0.875 | 0.700 |
| ACB | 0.854 | 0.864 | 0.932 | 0.872 |

Discriminant validity was also assessed by the Fornell–Larcker criterion [54,59]. Table 7 shows that the square root of the AVE for the construct in the diagonal was greater than the inner-construct correlation, which ranges from 0.934 to 0.548. The test result of the current study may therefore imply the strong reliability of all items.

**Table 7.** Fornell–Larcker Criterion. Note: Value in diagonal represents the square-root of AVE.

|  | ACB | AT | INTF | PBC | SN |
|---|---|---|---|---|---|
| ACB | 0.934 |  |  |  |  |
| AT | 0.835 | 0.874 |  |  |  |
| INTF | 0.725 | 0.746 | 0.836 |  |  |
| PBC | 0.315 | 0.263 | 0.197 | 0.548 |  |
| SN | 0.601 | 0.684 | 0.639 | 0.138 | 0.702 |

*4.6. Structural Model Analysis*

The structural or inner model of the research was tested with a standard algorithm. The result revealed that there is significance in the prediction of the construct in the research model. According to the 5000 resampling test, except for H3, H5, and H6, i.e., the perceived behavior control constructs, the data demonstrates positive and significant results for the rest of the hypotheses tests (H1, H2, H4, and H7) for the formalization of subsequent land transactions, Table 8.

**Table 8.** Mean, STDEV, T-Values, *p*-Values, R2.

|  | β | STDEV | T Statistics | *p* Values | 2.5% | 97.5% |
|---|---|---|---|---|---|---|
| AT-> INTF | 0.578 | 0.070 | 8.257 | 0.000 | 0.440 | 0.703 |
| AT -> PBC | 0.263 | 0.291 | 0.903 | 0.367 | −0.382 | 0.392 |
| AT -> SN | 0.684 | 0.034 | 20.182 | 0.000 | 0.624 | 0.753 |
| INTF -> ACB | 0.689 | 0.042 | 16.420 | 0.000 | 0.603 | 0.762 |
| PBC -> ACB | 0.179 | 0.187 | 0.956 | 0.340 | −0.269 | 0.268 |
| PBC -> INTF | 0.013 | 0.054 | 0.237 | 0.813 | −0.108 | 0.103 |
| SN -> INTF | 0.241 | 0.079 | 3.036 | 0.003 | 0.091 | 0.399 |
|  | $R^2$ |  |  |  |  |  |
| ACB | 0.556 |  |  |  |  |  |
| INTF | 0.587 |  |  |  |  |  |
| PBC | 0.069 |  |  |  |  |  |
| SN | 0.468 |  |  |  |  |  |
| AT | 0.000 |  |  |  |  |  |

According to the rule, the *t*-values of AT-> INTF, AT -> SN, INTF -> ACB, and SN -> INTF are well above 1.96, with $p = 0.000$ showing significant relationships. Whereas the t-values of AT -> PBC, PBC -> ACB, and PBC -> INTF are all below 1.96, with $p > 0.05$ indicating insignificant relationships. The results revealed that, except for the perceived behavior control constructs, there is significance in the prediction of the construct in the research model. Therefore, four of the seven hypotheses' tests have a positive and significant influence on the intentions and actions of the formalization of subsequent land transactions by the land rights holder. These are H1, H2, H4, and H7. The detailed analysis of these relationships is presented as follows.

Firstly, H1 evaluates whether the attitudes of land rights holders towards the value of land registration have a significant influence on their intentions to formalize subsequent land transactions. The result revealed a positive and significant influence on their intentions to formalize subsequent land transactions (β = 0.578, *t* = 8.257, *p* = 0.000), Table 8. Hence, H1 was fully supported. The result shows that 83.5% of the actual behavior of land rights holders' in terms of formalization was explained by attitudinal variables of the landholders towards the value of land registration. The attribute variables of attitudes include *secure*

*tenure* (AT1: 87.9%), *rights to fair compensation* (AT2: 92.4%), *access to individual credit* (AT3: 86.9%), *defense of boundary encroachment* (AT4: 88.6%), and *enforcement of rental contract* (AT5: 80.9%), Table 5 and Figure 5.

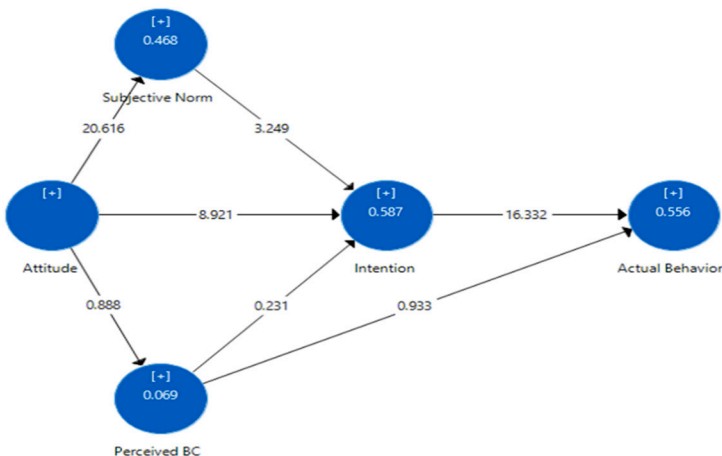

**Figure 5.** Measurement and structure equation model results of the research model.

Secondly, H2 estimates whether attitudes have a significant influence on the subjective norms of land rights holders' formalizing subsequent land transactions. The result revealed that the attitudes of land rights holders towards the value of land registration have a positive and significant impact on subjective norms ($\beta$ = 0.684, $t$ = 20.182, $p$ = 0.000), Table 8. Hence, H2 was also fully supported. Respondents admitted that 68.4% of attitudinal variables of the landholders influence the relationships with subjective norms, Table 7. The individual cross-loading of attitude variables to subjective norms ranges between 54.7 (AT3) and 65.4% (AT4), which shows above moderate relationships, Table 5.

Thirdly, H4 evaluates whether the subjective norms have a significant influence on land rights holders' intentions to formalize subsequent land transactions. The result revealed that subjective norms have a positive and significant impact on land rights holders' intentions to formalize subsequent land transactions ($\beta$ = 0.240, $t$ = 3.036, $p$ = 0.003), Table 8. Hence, H4 was also fully supported. In Table 7, respondents disclosed that 63.9% of the land rights holders' intentions to formalize subsequent land transactions was explained by the subjective norms attributable to the variables, see Figure 5 too. The attributable variables of the subjective norms include SN2—*most people who are important to land rights holders approve the formalization of all subsequent land rights transactions that I have made within six months* (36.5%), SN3—*spouse encourages the formalization of all subsequent land rights transactions that I made within six months* (48.1%), SN4—*few relatives (brother, sister, uncle, aunt, etc.) discourage the formalization of all subsequent land rights transactions that I made within six months* (20.2%), and SN5—*reporting and formalization made by a land rights holder would encourage other land rights holders to do the same within six months* (54.6%), Table 5.

Last but not least, H7 evaluates whether the intentions to formalize subsequent land transactions have a significant influence on the act of formalizing subsequent land transactions by the holders of the land rights. The result revealed that land rights holders' intentions to formalize have a positive and significant impact on the act of formalizing subsequent land transactions ($\beta$ = 0.689, $t$ = 16.420, $p$ = 0.000), Table 8. Hence, H7 was also fully supported. Respondents admitted that 72.5% of the actual behavior of land rights holders' formalization was explained by intention variables. The attribute variables of intentions include *a willingness to formalize subsequent land transaction within six months* (INTF1: 66.3%), *a willingness to pay for formalizing subsequent transaction costs in the next six months* (NTTR2: 55.2%), and *a willingness to participate in the private service provision scheme related to formalizing subsequent land transactions in the next six months* (INTF3: 59.7%), Table 5 and Figure 5.

In summary, subjective norms accounted for 46.8 percent of the variance in explaining intentions to formalize subsequent land transactions, Figure 5. Whereas, perceived behavioral control accounted for only 6.9 percent of the variance explaining intentions to formalize. On the other hand, intentions to formalize subsequent land transactions accounted for 58.7 percent of the variance explaining the actual behavior. Finally, the current model explained that 55.6 percent of the variance accounted for the act of formalizing subsequent land transactions by the holders of the land rights, which provides a substantial explanatory power and predictive capability of the current model.

Furthermore, the model fit was assessed using SRMR. The value of SRMR was 0.083, this is well below the required value of 0.10, indicating an acceptable model fit [55]. Further assessment of the validity of fit hypothesis is tested to ascertain the significance of the relationship, Table 9. The studied 5000 resamples also generate a 95% confidence interval, and hypotheses testing results are summarized in Table 9. A confidence interval that is not equal to zero indicates a significant relationship. Hence, the current models 95% interval shows 0.066 and indicates a significant relationship.

**Table 9.** Model fit analysis of the proposed model.

| | Original Sample (O) | Sample Mean (M) | 95% | 99% | Original Sample (O) | Sample Mean (M) | 95% | 99% |
|---|---|---|---|---|---|---|---|---|
| Saturated Model | 0.083 | 0.057 | 0.066 | 0.070 | 1.184 | 0.565 | 0.741 | 0.847 |
| Estimated Model | 0.101 | 0.061 | 0.070 | 0.075 | 1.729 | 0.647 | 0.847 | 0.973 |

### 4.7. Mediation Analysis

Mediation analysis was performed to assess the mediating roles of SN, PBC, and INTF on actual behavior outcomes. As can be seen from Table 10, SN was found to be significantly mediating the relationship between attitudes and intentions to formalize subsequent land transactions ($\beta$ = 0.165, $t$ = 3.013, and $p$ = 0.003), i.e., specific indirect effect. Similarly, INTF was found to significantly mediate the relationship between subjective norms and the actual behavior of land rights holders in formalizing subsequent land transactions ($\beta$ = 0.166, $t$ = 2.888, and $p$ = 0.004). Moreover, INTF was found to be significantly mediating between attitudes and the act of formalizing subsequent land transactions ($\beta$ = 0.398, $t$ = 7.225, and $p$ = 0.000). On the other hand, the results reveal an insignificant mediating role of PBC ($p$ > 0.05) between the attitudes and intentions to formalize subsequent land transactions by the holders of the land rights ($\beta$ = 0.003, $t$ = 0.231, $p$ = 0.818). Similarly, the result revealed that the same insignificant mediating role of INTF between the perceived behavioral control and the act of formalization by the land rights holders ($\beta$ = 0.009, $t$ = 0.233, $p$ = 0.816), Table 10.

**Table 10.** Total, specific, and indirect effect between the independent variables and dependent variables. T represents the t-statistics while Sig represents the significance or p value in the research model.

| | Total Effect | T | Sig | Total Indirect | Effect | T | Sig | Specific Indirect Effects | Effect | T | Sig |
|---|---|---|---|---|---|---|---|---|---|---|---|
| AT -> ACB | 0.561 | 14.007 | 0.000 | AT -> ACB | 0.561 | 14.007 | 0.000 | PBC -> INTF -> ACB | 0.009 | 0.233 | 0.816 |
| | | | | AT -> INTF | 0.168 | 2.912 | 0.004 | SN -> INTF -> ACB | 0.166 | 2.888 | 0.004 |
| | | | | PBC -> ACB | 0.009 | 0.233 | 0.816 | AT -> INTF -> ACB | 0.398 | 7.225 | 0.000 |
| | | | | SN -> ACB | 0.166 | 2.888 | 0.004 | AT -> SN -> INTF | 0.165 | 3.013 | 0.003 |
| | | | | | | | | AT -> PBC -> INTF | 0.003 | 0.231 | 0.818 |
| | | | | | | | | AT -> PBC -> INTF -> ACB | 0.002 | 0.229 | 0.819 |
| | | | | | | | | AT -> SN -> INTF -> ACB | 0.114 | 2.852 | 0.005 |
| | | | | | | | | AT -> PBC -> ACB | 0.047 | 2.084 | 0.038 |

## 5. Discussion

Both the FLLC and SLLC land registration programs were implemented in the study area over the past one and a half decades. This is part of the national land registration process. The policy drive behind this program is to improve the security of land and resource tenure to incentivize landholders to invest in sustainable land use practices that, in turn, would bring about sustainable livelihoods and environmental outcomes. Despite the two stages of successful large-scale land registration programs, the number of subsequent land transactions registered and updated in the land registers is less than 1% of the total number of parcels [21]. Thus, regardless of the success of first-time registration, the volume of formalizing (registering) subsequent land transactions is still low and threatens the accuracy, currency, and completeness of the information in the land registers across the country.

The current study, therefore, tries to explore whether land rights holders' behavior is driving the lack of updating of the land register. If so, what factors influence land rights holders' intentions and the act of formalizing land transactions? What must be carried out to ensure the function and sustainability of the land information system in the country? These are topical and valid research questions that this study tries to answer.

This study demonstrated that the theory of planned behavior (TPB) can be employed to deepen our understanding of land rights holders' formalization of subsequent land transactions. Among other issues, the authors considered and explored the influencing factors of land transaction formalization and their relationships, including attitudes, subjective norms, and perceived behavioral control, as antecedents to landholders' intentions to the act of formalizing land transactions in Ethiopia. The authors also considered the role of demographic and socioeconomic background factors, such as institutional policies and regulations, societal customs and values, and demographic characteristics. Below is the discussion of the results in detail.

Firstly, based on the structural model (inner model), the paper examined the relationship between land rights holders' attitudes towards the values of land registration and their intentions to formalize subsequent land transactions. The result indicates that there is a positive and significant relationship between attitudes towards, and intentions to formalize, subsequent land transactions. Hence, it can be inferred that land rights holders relate their beliefs on the value of land registration with their intentions to formalize land transactions that they have made, after first-time land certificate issuance.

The attitudes constructs reflect whether the land rights holder is in favor of formalizing subsequent land transactions, or the land rights holder's beliefs about the consequences of registering one's land use rights in the formal land register. These values are influenced by the outcomes of land registration, including secure tenure, transferability of rights (inheritance or donation, rights to fair compensation, access to credit, and defense against boundary encroachment). In addition, the perceived benefits of land registration relate to the landholders' perceptions about the impact of land registration on the enforcement of other social contracts including a lease, rent, or joint venture.

The study suggests that the prediction tends to be quite accurate in its aggregate form or the macro level, but not at the individual level, between intentions to, and the actual behavior of, completing the formalization of subsequent land transactions. This means that the correlation between the intentions to get subsequent land transactions formalized, and the actual behavior of the same, in the coming six months, was 93.2 percent at the aggregate level, but ranges from 26.8 percent to 94.2 percent at the individual level, Tables 3 and 5. This result indicates a strong intention to formalize but less actual completed subsequent land transactions at the individual level. This result is also consistent with that of the qualitative findings of this research carried out through KII. Some of the reasons justifying the gap between strong intentions but less actual formalized land transactions may be inadequate awareness among land rights holders. Moreover, some landholders feel transactions were among families and relatives, so there is no need to register, and did not feel any risk in prior transactions, so there is no need to register. This result is also quite

similar to the results in other fields of study, such as in demography, on the intended and actual family size [60].

Secondly, subjective norms refer to how a land rights holder feels social pressure to formalize the subsequent land transaction. The results of this study are consistent with other studies dealing with different behavior (voting, reducing household waste, and energy consumption), as summarized by Barbera and Ajzen [40]. Most importantly, as found by Edwin [61] and Kingwill [62], social property regimes display characteristics that are not easily registered as people to property relations hinge on localized customs and social networks that defy the formal registration of all transactions. Abubakari et al. [11] also found that the registration of subsequent land transactions also drives from and is underpinned by socio-political struggles and strategies, for example, across gender and between hierarchies of those within land governance structures.

The current study also revealed that subjective norms have a positive and significant impact on land rights holders' intentions to formalize subsequent land transactions. Updating the land registers is one of the key functions of the woreda land offices. Depending on the types of land transactions, land rights holders are legally obliged to formalize subsequent land transactions that they have made within a certain period. However, their intentions to formalize subsequent land transactions are influenced by the social customs and values of the communities towards registration. The better the customs and values of the communities towards land registration, the stronger the intentions of landholders to report and formalize subsequent land transactions.

Based on the result of this study, it can be concluded that normative beliefs of land rights holders result in perceived social pressure or subjective norms. The subjective norms, in turn, influence the intentions of land rights holders to formalize subsequent land transactions. Therefore, the formalization of subsequent land transactions need to be seen in the context of social norms and customs. This finding is similar with what Abubakari et al. found in their recent work in Ghana [63]. They found that the decision to register inherited land is a co-produced agency of different norms and is influenced by multiple normative, official, social, and practical frames [63,64].

Thirdly, regarding perceived behavioral control, landholders' intentions to formalize subsequent land transactions could be determined by their control beliefs about the power of both situational and internal factors to inhibit or facilitate performing the behavior. The results show that perceived behavioral control has a weak predictive relation to intentions to formalize subsequent land transactions. This result is somehow counterintuitive to the individual level formalization behavior. However, information from the key informant interview has strongly supported this finding as to why such weak variance is accounted for by perceived behavioral control in the current model. This may be because land rights holders in the study area consider land as the ultimate resource of their livelihoods, and privileges of identity in their community. Hence, landholders are ready to pay any cost—including life to defend their land rights. That is why most respondents invariably responded to the measurement items related to perceived behavioral control. This indicates that we need to be careful when interpreting the coefficients of the model.

Barbera and Ajzen [40] advise that the question of actual control has little bearing on the application of the TPB to predict intentions to attain a behavioral goal [40]. Barry [65] also found that, in volatile situations, investigations should include comparative cases of actual behavior of the cadastral system. In the present study, a behavioral goal could be exclusive enjoyment of one's landholding rights, but the intention here is to formalize subsequent land transactions after a first-time land registration. The study result revealed that some regulatory and administrative barriers, such as rules on minimum parcel/holding size, may hinder the registration of subsequent land transactions [57,58].

Contrarily, health condition, economic, and marital status can define actual control, which, if favorable, will enable land rights holders to act on their intentions to formalize subsequent land transactions or, if unfavorable, will make it difficult for them to get

subsequent land transactions formalized, despite their intentions to do so. To illustrate it further, a land rights holder who intends to formalize subsequent land transactions may have a good health condition, emotional status, be able to pay a registration fee, knows the procedure of registration, and feels that registration is accessible, or the procedure is easy and affordable, and thus exhibits control over the required formalization behavior. Yet, he/she may fail to attain his/her goal if his/her spouse is unwilling to do so, or if registration is impossible due to court cases/injunctions or some other regulatory and administrative requirements. The recent literature in the field have well documented considerations to overcome land administration maintenance challenges and recommended solutions and specific examples in the context of the overall framework for effective land administration establishment, maintenance, and upgrading [3,12,66].

## 6. Conclusions and Recommendations

The purpose of the present study was to examine the factors that influence landholders' intentions and the act of formalizing rural land transactions after a first-time registration. Using the theory of planned behavior (TPB), this study has tested the hypotheses that influence land rights holders' intentions to formalize land transactions. The study demonstrated that the application of the TPB can be employed to deepen our understanding of land rights holders' intentions to formalize land transactions. By examining behavioral, normative, and control beliefs about getting subsequent land transactions formalized, the study identified important policy and operational considerations that influence this behavior. The structural equation model (tested hypotheses) findings are also substantiated by the descriptive and inferential statistics findings. Further, the findings are also supported by qualitative data analysis and findings obtained through key informant interviews.

Based on the results of this study, it can be concluded that landholders' attitudes to land registration benefits and their perceived normative beliefs (subjective norms) can influence their intentions and actual practices of formalizing land transactions. An implication of this is the possibility that affects the currency, accuracy, and completeness of information in the land register. Participants in the study admitted that the two stages of the land registration program brought about positive development outcomes, including improved secure tenure and opened up new opportunities such as SLLC-linked individual loans. However, the longer-term functionality and sustainability of the land registration system is essential if the land register is to be kept up-to-date after first-time registration. The land registration system should be able to facilitate subsequent land transaction formalization. Land rights holders need to be aware of, and believe that, land transactions need to be registered in the formal land registration system. Since the SLLC program was completed a couple of years ago in the study area, local government capacity needs to shift gears towards the maintenance and updating of the land register. The attention given to land transaction formalization will have far reaching consequences, not only to the land registration system sustainability, but also to the overall land governance system effectiveness in the country.

Therefore, this study underscored that facilitating landholders' attitudes to the formalization of land transactions and helping them to overcome hurdles facing subjective norms are important issues for the functionality and sustainability of the land registration system in Ethiopia. The study findings recommended facilitating the behavioral changes of landholders to transform their strong intentions into actual practice. Policymakers should develop and implement an innovative information value creation strategy, including landholder-oriented services that incentivize the formalization of land transactions and help landholders overcome hurdles created by subjective norms.

The strategy may include continuing regular awareness-raising campaigns, familiarizing some incentive mechanisms such as making service centers closer to the communities for remote and inaccessible kebeles, and introducing a private service providers' scheme for selected transactions such as land rentals. Regular monitoring and incentivizing mechanisms for registering land transactions at the woreda and kebele level would also be

considered to improve the formal registration of land transactions. This could establish a system where services and service delivery centers are customized to local contexts, and are simple, accessible, affordable, accurate, and quick. Moreover, some regulatory and administrative document requirements should be eased, such as removing marriage certificate submission as a supporting document for registering subsequent land transactions. Such requirements disincentive landholders to register land transactions since marriage certificates are not a customary practice among the rural societies in the country.

The findings reported here shed light on the importance of updating the land register through formalizing land transactions and factors influencing landholders' intentions to do so. The present study is one of the few studies that attempted to thoroughly examine factors that influence landholders' intentions of formalizing land transactions after a first-time land registration in Ethiopia. As with any development interventions, land administration programs are a planned public policy and development intervention. Hence, future research can use intentions that help to predict the performance of desired outcomes in the complex land administration and management public policy reforms. In addition, future research can consider additional predictors such as trust and the legitimacy of the registration system that drive landholder's behavior and test it as mediators between perceived behavioral control and intentions that could increase the explained variance. Moreover, future research can consider the effects of interactions of variables on intentions in the current model. The interactions can be between subjective norms and perceived behavioral control, or other additional predictors suitable to the field, and their effect on intentions and actual behavior.

**Author Contributions:** Conceptualization, S.A.A., F.S.W. and T.T.N.; methodology, S.A.A.; software, S.A.A.; validation, S.A.A., F.S.W. and T.T.N.; formal analysis, S.A.A.; investigation, S.A.A.; resources, S.A.A., F.S.W. and T.T.N.; data curation, S.A.A.; writing—original draft preparation, S.A.A.; writing—review and editing, S.A.A., F.S.W. and T.T.N.; visualization, S.A.A.; supervision, F.S.W. and T.T.N.; project administration, Kelly Robbins; funding acquisition, S.A.A., F.S.W. and T.T.N. All authors have read and agreed to the published version of the manuscript.

**Funding:** The research was supported by the Partnerships for Enhanced Engagement in Research (PEER) Program, financed by the U.S. Agency for International Development (USAID) and administered by the U.S. National Academy of Sciences (NAS) under cooperative agreement AID-OAA-A-11-00012. The authors highly acknowledge and appreciate this generous financial support, without which this research could not have been realized.

**Institutional Review Board Statement:** Not applicable.

**Informed Consent Statement:** Informed consent was obtained from all subjects involved in the study.

**Data Availability Statement:** Not applicable.

**Acknowledgments:** Our heartfelt gratitude goes to Dilu Shaleka, former Dean of College of Development Studies (CDS) of Addis Ababa University and Tesfay Zeleke who replaced him, and their finance team who have been supportive of all administrative matters. The authors would also like to extend our thanks to Kelly Robbins, Senior Program Officer, of the National Academies of Sciences, Engineering, and Medicine (NAS), who administers the PEER grant funds, for her unwavering support for our research work. Thanks also goes to Negese Negash, Adugna Asrat, Alemenesh Tesfaw, Dawit Gesit, Eshetu Ayele, Genet Cherinet, Kebede Girma, and Mekasha Kinfe from Basona Worena woreda land office for their excellent contribution to the data collection process to this study without their persistent commitment the data collection could not succeed.

**Conflicts of Interest:** The authors declare no conflict of interest.

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
