# Peer review of "Factors Influencing the Formalization of Rural Land Transactions in Ethiopia: A Theory of Planned Behavior Approach"

_land, doi:10.3390/land11050633_

Round 1

Reviewer 1 Report

The submitted paper describes the application of the theory of planned behavior (TPB) to investigate intentions to formalize land transactions and the lack of update existing land records in Ethiopia.

The paper touches upon a very relevant issue around maintenance/updating of land administration systems after initial land recordation (first-time land certification). It presents a good overview of incentives around formalization of land transactions. The general impression of the reviewer is that this submitted paper fits very well in this journal and can be published in this scientific journal. The hypothesis raised were sufficiently answered with the methods used in this study. Further the references used are appropriate, current and use was made of relevant frameworks (e.g. FELA). The manuscript is clear and well-structured. The description of the methodology is very extensive though because of the complexity of the analysis this extensive description seemed relevant.

Especially the discussion holds very relevant observations and the arguments presented are consistent and are basically reconfirming the importance of the research as presented in this manuscript. Interesting was also the investigation on how much money the landholders are willing to pay per SLLC in relation to the value. Congratulations to the authors and more research like this is needed to redirect the focus also to maintenance issues in the land administration domain.

Author Response

Dear Reviewer,

Many thanks for your detail review and positive feedback, very much appreciated. Based on all reviewers comments and suggestions, a revised version of the manuscript is uploaded for your further review.

Once again thank you so much for the opportunity and your scientific feedback.

Best Regards

Shewakena on behalf of all authors

Reviewer 2 Report

Summary

The authors investigate the factors influencing the formalisation of subsequent land transactions in an Ethiopian district. Results of the surveys and key informant interviews document that attitudes and subjective norms positively contributes to the intension of land holders to update the land register in case of a subsequent land transaction.

General Comments

Many publications document the land registration processes in countries and identify the implementation of a land administration system as a big success. However, in many countries, less importance is given to the maintenance and updating of data resulting in outdated data bases. The current article investigates the attituded of farmers in their willingness to formalise subsequent land transaction and therefore the paper is an important contribution for a well-working the land administration sector. In addition, the topic of the article fits to the aims of the journal.

The structure of the paper is according to a scientific publication. The introduction gives a good insight into the topic being investigated. Chapter two provides the readers with the basics of the Theory of Planned Behaviour and its application for investigating land administration systems. Methods, results, discussions, and conclusions are the headings of the following chapters. As the content of these chapters is not always according to the headings, the paper has potential for improvement and has to be revised.

General suggestions for improvement:

  • Rename Chapter 3 to “Study Site and Methods”
  • In chapter 3, summarize paragraphs before sub-chapter 3.1 in a short paragraph and integrate text (line 282 to line 339) in a consolidated format to a new chapter 3.2 “Methods for data assessment and data analysis”.
  • Add final questionnaire as appendix to the paper
  • Shift description of validation methods (line 441 o 455) to new chapter 3.2, and adapt the presentation of the results in chapter 4
  • Shift the description of the SMA (line 487 to 501) to the new chapter 3.2 and adapt the presentation of results in chapter 4
  • Chapter “Discussion” has to be revised:
    • Discuss the results in accordance to the structure of chapter 4.
    • Avoid repetition of results.
    • Integrate comments of land administration experts given in the key informant interviews
    • Compare the results with findings of other studies
    • Delete text not focused to your findings (e.g. line 599 to 603)
  • Some recommendations and conclusions are given in the discussion part – shift this text to the final chapter, which can be headed by “Conclusions and recommendations”
  • Check references on completeness, correct format and correct citation, [e.g. 22, 25, 29, 49, 50, … ]

Additional, more detailed proposals for improvements

Line 190:     Substitute “on the other hand”, as you did not start with “on the one hand”

Line 276:     Renumber Figure 2 to Figure 1

Line 337:     Justify, why the tablet-based data collection is reducing data entry errors

Figure 2:      Improve figure: Add scale bar to the map of Amhara region and Basona worena woreda; Check grid labelling as these are cut or hidden behind the frame

Line 407f:    you selected seven kebeles, but you listed only 6 of them.

Line 422f:    Only a histogram can proof your finding (“the majority of the landholders are in their productive ages”), not the mean value!!! – Document also your understanding of “productive age”

Table 2:       Reduce number of decimal places

Line 631ff:   After “firstly” at least “secondly” is expected….

Line 715:     avoid “On the other hand” – see above

Author Response

Dear Reviewer,

Thank you so much for the thorough review, comments, and suggestions for our submitted manuscript with ID number land-1663239, highly appreciated it. We have tried to address all comments and suggestions adequately and made a revision to the manuscript. The revised manuscript is uploaded with response matrix for your further review and guidance.

Once again thank you for the opportunity and your scientific insights and positive feedback as well.

Best regards

Shewakena on behalf of the authors

Reviewer 3 Report

Having read this manuscript, I consider it to have merits for publication. However, I recommend improving the following aspects.

  • It is no news that there is relationship between behavior and development interventions. In this case, its difficult to grasp the qualitative dimension of the hypothesis tested as it appears a little over simplified. Behavior is a qualitative issue, and its relationship to land administration activities goes beyond mere test of significance. There are more issues related to culture, policy (and others) that can best be described.
  • The recommendation is also not very new. The authors argue that "future land administration and management research could widely apply TPB since a land administration intervention is a planned public policy development intervention." The essence of "participation" in land administration (LA) poject is usually to align the interest of communities with overall project or even have them drive the process. The cultural concerns are part of the TPB (in LA context). So the paper leaves me, wondering what its key novelty is.
  • This is obvious because the manuscript lacks a dedicated "land governance/policy implication" section that teases out the actual land administration implications of its outcome. This needs improving.

Author Response

Dear Reviewer,

Thank you so much for your scientific comments, feedback, and suggestion for our submitted manuscript with ID number land-1663239, highly appreciated. Considering all reviewers comments and feedback, we have made a revision to the manuscript and uploaded to the journal portal for your further review and guidance.

Firstly, we have noted and tried to address all your valuable comments in the revised version of the paper. It was commented that the qualitative dimension of the hypothesis tested as it appears a little over simplified. We fully agree that behavior is a complex and qualitative issue, particularly in the land administration domain. For instance, when we try to measure tenure security we are measuring the perception levels of people as a proxy measurement variables. This holds true for measuring or understanding landholder's behavior towards registering subsequent land transactions. Understanding this complexity we have tried to use both quantitative and qualitative methods in our research methods. We have integrated the comments and observations of our KIIs in qualitative form which was missed in the previous version of the paper. This helps us to substantiate the quantitative results of the paper. 

Secondly, it was mentioned that the cultural concerns are part of the TPB (in LA context). Confidently, this is also true, particularly the social norm construct concerns about custom and culture. However, TPB is beyond culture and does provide a structure for designing a study and it provides the class of data that should be collected and analyzed in quantitative manner to reach to understandable conclusions and recommendations.

Finally, as commented, we have noted that the recommendation section needs revision that highlight the land governance implications. The revision takes into account this serious setback of the previous version of the paper and we have made necessary changes and improvements.

Once again many thanks for the opportunity, your valuable insights, and positive feedback.

With Best Regards

Shewakena on authors behalf

Round 2

Reviewer 3 Report

The paper has addressed all potentially serious issues I could see in it. However, minor concerns such as the caption (title) can send an embarrassing message to readers.

The current title "Factors Determine Landholders’ Behavior towards Formalization of Subsequent Land Transactions in Ethiopia: A Theory of
Planned Behavior Approach" does not read smoothly in comprehension. The word "determine" either needs to be re-tweaked or it could be improved with an article such as "that" before it, then there is the word "towards" which reads like it is misplaced. The word "subsequent" also does not do much. I understand it is there to reflect the remnant (or phase) of the Ethiopian registration exercise being investigated but this is clear in the paper. If you must describe the "phase" then be specific about the official project title (e.g. Phase 2 or... whatever being used) or leave it out. The key message appears to be along the line of titles such as (by way of example):

(1) Using the theory of planned behavior to assess the behavioral factors that motivate landholders’ to formalize land transactions in Ethiopia

(2) Behavioral factors that determine landholders’ to formalize land transactions in Ethiopia from: a theory of planned behavior approach

(3) Or any others the authors could coin. But the title has to send a clear message about the content of the paper.

The authors should consider re-captioning the paper to reflect its key message.

Language: the paper could benefit from moderate edits to tease out some awkward sentences. -e.g.:

(1) "programs are a planned..." (line 1005). There several of this sort of expressions that relate to usage of singular and plural.

(2) "Recent literature well documented considerations..." (line 954). There are several of this sort of incomplete expressions all over the paper.

(3) "This result also consistent with that of the qualitative findings of this research done through KII." (lines 884-885). Did you see your use of "... qualitative findings of this research done through KII." Does that imply this research has been done before (qualitatively)? Not clear.

There are a lot of these expressions in the paper, This manuscript needs very tight editing to tease out these correct-reading sentences that in reality can be deemed wrong or unscientific.

Author Response

Dear Reviewer,

Many thanks again for recognizing our major modifications made against the first round of comments and your new suggestions for further consideration, much appreciated. Accordingly, we have made necessary revisions and resubmitted the revised version of the paper. The title of the paper is now modified to reflect your suggestion, intensive edits have been done, and conclusion and recommendation section improved. We hope you will find the current version qualified for publication given that we have addressed all comments and suggestions adequately.  

Once again many thanks for the opportunity and looking forward to your approval.

Best Regards

Shewakena on behalf of the authors

This manuscript is a resubmission of an earlier submission. The following is a list of the peer review reports and author responses from that submission.